# Process Optimization and Microstructure Analysis to Understand Laser Powder Bed Fusion of 316L Stainless Steel

Nathalia Diaz Vallejo [1], Cameron Lucas [2], Nicolas Ayers [1], Kevin Graydon [1], Holden Hyer [1,3] and Yongho Sohn [1,*]

1   Advanced Materials Processing and Analysis Center, Department of Materials Science and Engineering, University of Central Florida, Orlando, FL 32816, USA; nathalia.diaz@Knights.ucf.edu (N.D.V.); nicayers16@Knights.ucf.edu (N.A.); kevingraydon@knights.ucf.edu (K.G.); hyerhc@ornl.gov (H.H.)
2   Advanced Materials Processing and Analysis Center, Department of Mechanical Engineering, University of Central Florida, Orlando, FL 32816, USA; cameronlucas@Knights.ucf.edu
3   Now with Oak Ridge National Laboratory, Oak Ridge, TN 37831, USA
*   Correspondence: yongho.sohn@ucf.edu

**Abstract:** The microstructural development of 316L stainless steel (SS) was investigated over a wide range of systematically varied laser powder bed fusion (LPBF) parameters, such as laser power, scan speed, hatch spacing and volumetric energy density. Relative density, melt pool width and depth, and the size of sub-grain cellular structure were quantified and related to the temperature field estimated by Rosenthal solution. Use of volumetric energy density between 46 and 127 J/mm$^3$ produced nearly fully dense ($\geq$99.8%) samples, and this included the best parameter set: power = 200 W; scan speed = 800 mm/s; hatch spacing = 0.12 mm; slice thickness = 0.03; energy density = 69 J/mm$^3$). Cooling rate of $10^5$ to $10^7$ K/s was estimated base on the size of cellular structure within melt pools. Using the optimized LPBF parameters, the as-built 316L SS had, on average, yield strength of 563 MPa, Young's modulus of 179 GPa, tensile strength of 710 MPa, and 48% strain at failure.

**Keywords:** laser powder bed fusion; 316L stainless steel; melt pool dimension; microstructure

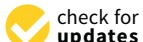



## 1. Introduction

Additive manufacturing (AM), commonly known as three-dimensional (3D) printing, is an emerging technology initially proposed for producing prototypes, but nowadays employed for building functional and structural components with complex geometry from a 3D model [1,2]. For metal AM, fundamental knowledge from welding metallurgy can be used to better understand and improve the additive manufacturing techniques [3]. For example, the results from the studies of dissimilar metal welds and joints [4,5], could be useful to understand the melting and solidification phenomena that could help to locally tailor alloy composition and properties in additive manufacturing. In laser powder bed fusion (LPBF), powder feedstock is spread over as a powder bed layer and selectively melted by a laser source to build a component layer by layer [6–9]. A number of commercial alloys have been successfully used to manufacture engineering components by LPBF, including those based on titanium [10–13], iron [14–16], aluminum [17–19], or nickel-based superalloys [20,21], etc.

Austenitic stainless steels commonly processed in LPBF systems are grade 304L [22,23] and 316L [14,15,24–26]. The austenitic 316L stainless steel (316L SS) with chromium-nickel-molybenum alloying additions, has a carbon content lower than 0.03 wt. % and good weldability. Excellent corrosion resistance of 316L SS is attributed to the formation of a stable passive chromium-rich oxide film on the surface, due to the presence of chromium (Cr > 16 wt. %). LPBF 316L SS has been reported to have a superior corrosion resistance from their traditionally manufactured counterparts [24–26], due to the absence of MnS inclusions, which has been reported to be associated with the initiation of pitting corrosion [27]. However, LPBF 316L SS has been shown to have a lower re-passivation

potential [24,25,28], possibly due to the presence of internal porosities and the inhomogeneous microstructure. Given its mechanical, weldability, and corrosion resistance, 316L is of great interest to numerous applications including marine [29], biomedical equipment [30] and fuel cells [31].

In LPBF the most influential processing parameters on the quality of the parts are the laser power (P), the scan speed of the laser (v), the distance between consecutive laser scans, known as hatch spacing (h), and the powder layer or powder bed thickness, commonly referred to as slice thickness (t) [32–37]. These parameters play a vital role on the microstructure, physical and mechanical properties of the parts produced by LPBF, and their optimization is the key to minimize defects and stress risers, such as lack of fusion, keyholing, the balling effect, internal cracks, changes in chemical composition and thermal stress.

The total energy input per unit volume or volumetric energy density [38] have been used as an approach in order to simplify the LPBF parameters, and it combines the aforementioned parameters as:

$$\text{Volumetric energy density} = \frac{P}{v \cdot h \cdot t} \qquad (1)$$

Cherry et al. [35] reported minimum porosity of 0.38% obtained at an energy density of 104.52 J/mm$^3$. Most of the studies since then show that the energy density to achieve nearly full dense 316L SS is approximately 100 J/mm$^3$ [36,37]. Despite the significant progress in producing a high-density material [35,39–41], porosities in LPBF might occur under two completely different conditions: insufficient volumetric energy density yielded to lack of fusion [42,43], and excessive volumetric energy density produced porosity that can be attributed to the occurrence of keyhole-mode [44]. Mechanical properties of the LPBF 316L SS are generally considered superior to their traditionally manufactured counterparts [41,45–48], and its dependency with the processing parameters has been confirmed in several studies [36,46,49–52]. Notably, Suryawanshi et al. [47] reported an increase in yield strength by 60% for the LPBF 316L SS. Sun et al. also [46] showed that the tensile strength and ductility of LPBF 316L SS can be enhanced by 16% and 40%, respectively, through the manipulation of the processing parameters and thus the crystallographic texture. The substantial enhancement in yield strength has been attributed to the marked refinement in the micro-structure of 316L SS, which is a result of the high cooling rates (~$10^5$ to $10^8$ K/s) achieved during the LPBF processing [15,45,53–56].

The aim of this study is to provide a comprehensive understanding of the effect of LPBF processing parameters, namely the laser power, scan speed, hatch spacing, independently and systematically varied to extreme magnitudes, on densification, melt pool characteristics and microstructure development in 316L SS based on the volumetric energy density approach. Optimum LPBF parameters were also established based on the volumetric energy density. Melt-pool width and depth were measured from experiments and compared to those estimated by the analytical Rosenthal solution. Cooling rates were estimated based on microstructural quantification and mechanical properties in tension were determined for the as-built 316L SS specimens produced by optimum LPBF parameter set. The results reported contribute to a better understanding of LPBF process for SS316 by providing quantified microstructural data, which would also be valuable for simulations (e.g., mechanism-based or data-driven machine learning). Moreover, in general, findings from this study would contribute to the geometry and composition dependent LPBF optimization of ferrous alloys.

## 2. Materials and Methods

### 2.1. 316L Stainless Steel Powders

The starting material used in this study was gas atomized 316L SS powders purchased from SLM Solutions (SLM Solutions Group AG, Lübeck, Germany). Prior to LPBF, the powder size distribution was determined using a Beckman-Coulter (Beckman Coulter, Inc.,

Brea, CA, USA) LS 13 320 laser diffraction particle size analyzer. Powder morphology and cross-sections were examined with the field emission scanning electron microscopy (FE-SEM) Zeiss Ultra-55 (Carl Zeiss AG, Jena, Germany), equipped with an X-ray energy dispersive spectroscopy (XEDS) operating at an accelerating voltage of 20 kV. Standardless semi-quantitative analyses from XEDS data were carried out using Noran System 7 Version 3.0 software (Thermo Fisher Scientific Inc., Madison, WI, USA) for the estimation of compositions.

### 2.2. Laser Powder Bed Fusion

An LPBF system, SLM$^®$ 125HL (SLM Solutions Group AG, Lübeck, Germany), equipped with a single 1070 nm wavelength IPG fiber laser of up to 400 W laser power with a beam diameter of about 70 μm was employed to produce cylindrical samples (12 mm in height and 6 mm in diameter). Build direction was parallel to the height of the cylindrical samples. Preheating of the build plate was set at 100 °C and Ar was used to maintain the insert atmosphere by keeping the $O_2$ content below 0.2% during LPBF.

The processing parameters, laser power, scan speed and hatch spacing, were varied independently with due consideration for normalized volumetric energy density (Equation (1)), as listed in Table 1. The following conditions were held constant throughout this study: 0.03 mm layer thickness (*t*), 10 mm stripe width, 0.08 mm stripe overlap, 67° scanning rotation between subsequent layers and a stripe scanning strategy. The contouring, up-skin, and down-skin parameters were all deactivated so that only the parameter variation listed in Table 1 was varied throughout the investigation.

The optimized parameter set based on specification by SLM Solutions Group AG has a laser power, scan speed, hatch spacing, and slice thickness of 200 W, 800 mm/s, 0.12 mm, and 0.03 mm, respectively. This corresponds to an energy density of ~69 J/mm$^3$, and is included in Table 1. The initial layer of the build was set such that cross-sectional microscopy would allow for the measurement of melt pool width and depth using the final, top LPBF layer, without any repeated laser melting.

**Table 1.** Laser powder bed fusion parameters examined in this study for 316L stainless steel.

| Series | Power (W) | Scan Speed (mm/s) | Slice Thickness (mm) | Hatch Distance (mm) | Energy Density (J/mm$^3$) | Relative Density (%) Measured by Image Analysis * |
|---|---|---|---|---|---|---|
| I | 125 | 100 | 0.03 | 0.12 | 347.2 | 96.35 ± 0.79 |
| | | 200 | | | 173.6 | 96.79 ± 1.35 |
| | | 400 | | | 86.8 | 99.09 ± 0.28 |
| | | 600 | | | 57.9 | 99.90 ± 0.08 |
| | | 800 | | | 43.4 | 99.51 ± 0.30 |
| | 200 | 200 | | | 277.8 | 98.35 ± 0.53 |
| | | 400 | | | 138.9 | 99.49 ± 0.27 |
| | | 600 | | | 92.6 | 99.92 ± 0.05 |
| | | 800 | | | 69.4 | 99.89 ± 0.06 |
| | | 1000 | | | 55.6 | 99.88 ± 0.07 |
| | | 1200 | | | 46.3 | 99.83 ± 0.05 |
| | | 1400 | | | 39.7 | 99.44 ± 0.20 |
| | | 1800 | | | 30.9 | 96.60 ± 1.74 |
| | | 2200 | | | 25.3 | 94.23 ± 1.00 |
| | | 2600 | | | 21.4 | 90.22 ± 3.17 |
| | 275 | 400 | | | 191.0 | 98.74 ± 0.64 |
| | | 600 | | | 127.3 | 99.9 ± 0.07 |
| | | 800 | | | 95.5 | 99.91 ± 0.14 |
| | | 1000 | | | 76.4 | 99.98 ± 0.01 |
| | | 1200 | | | 63.7 | 99.77 ± 0.10 |
| | | 1400 | | | 54.6 | 99.87 ± 0.06 |
| | | 1800 | | | 42.4 | 99.62 ± 0.14 |
| | | 2200 | | | 34.7 | 99.09 ± 0.41 |
| | | 2600 | | | 29.4 | 96.55 ± 0.59 |
| | | 3000 | | | 25.5 | 93.52 ± 1.62 |
| | 350 | 600 | | | 162.0 | 99.74 ± 0.08 |
| | | 800 | | | 121.5 | 99.93 ± 0.05 |
| | | 1000 | | | 97.2 | 99.90 ± 0.03 |
| | | 1200 | | | 81.0 | 99.98 ± 0.02 |
| | | 1400 | | | 69.4 | 99.78 ± 0.23 |
| | | 1800 | | | 54.0 | 99.71 ± 0.18 |
| | | 2200 | | | 44.1 | 99.46 ± 0.32 |
| | | 2600 | | | 37.4 | 95.09 ± 2.00 |
| | | 3000 | | | 32.4 | 94.40 ± 1.47 |
| | | 3400 | | | 28.6 | 94.90 ± 1.32 |
| II | 200 | 800 | 0.03 | 0.08 | 104.2 | 99.98 ± 0.03 |
| | | | | 0.1 | 83.3 | 99.99 ± 0.01 |
| | | | | 0.12 | 69.4 | 99.97 ± 0.02 |
| | | | | 0.14 | 59.5 | 99.91 ± 0.04 |
| | | | | 0.16 | 52.1 | 99.84 ± 0.07 |

* Average ± Standard deviation.

### 2.3. Characterization of Microstructure and Mechanical Behavior

Each cylindrical sample was cross-sectioned both parallel and perpendicular to the build direction, and metallographically prepared with SiC grinding papers and diamond paste polishing, with a final finish of 0.05 mm using colloidal silica. The polished cross-sections were examined by optical microscopy using Nikon Metaphot (Nikon Metrology Inc, Tokyo, Japan). ImageJ [57] (National Institute of Health, Bethesda, MD, USA) was employed to quantitatively determine the amount of flaw/porosity and their circularity,

based on five cross-sectional micrographs (100× magnification) taken before etching at random locations from each cross-section. For a given sample, the amount of porosity/flaw observed did not vary significantly between the cross-sectional planes in build direction and the normal, so only the results from the cross-section parallel to the build direction were reported and analyzed. All cross sections were then etched by immersion for approximately 40 to 60 s in the mixed acids reagent consisting of hydrochloric acid: acetic acid: nitric acid in 3:2:1 volume ratio for detailed microstructural analyses.

To assess the mechanical behavior of as-built LPBF 316L SS, tension testing was carried out using the traditional dog-bone specimens with a gauge length of 25 mm in accordance with tolerances described in ASTM E8/E8M. Three tensile specimens were produced using the optimized parameter set identified in this study (power = 200 W; scan speed = 800 mm/s; hatch spacing = 0.12 mm; slice thickness = 0.03; energy density = 69.4 J/mm$^3$).

## 3. Estimation of Melt Pool Dimensions

The three-dimensional temperature field and melt-pool dimensions were estimated using a simplified analytical solution for a moving point heat source introduced by Rosenthal [58], expressed by:

$$T = T_0 + \frac{Q}{4\pi k} \frac{\exp\left(\frac{-vR}{\alpha}\right) \cdot \exp\left(\frac{-vx}{\alpha}\right)}{R} \tag{2}$$

where $T_0$ is the temperature of the powder bed taken here as 100 °C of the build plate, $Q$ is the laser power absorbed, $k$ is the thermal conductivity, $a$ is the thermal diffusivity, $v$ is the scan speed of the laser source moving along $y$ direction, and $R$ is the distance to the edge of the melt pool from the laser point source defined by:

$$R = \left(x^2 + y + z^2\right)^{1/2} \tag{3}$$

where $z$ is defined as the direction of the incident laser and $y$ is the direction of the laser can. The absorbed power (i.e., effective power input) was estimated as:

$$Q = A \times P \tag{4}$$

where A is the laser absorptivity and P is the actual laser power employed by LPBF.

This model based on Rosenthal solution was examined using MATLAB® to estimate the temperature field within the XZ plane along the weld line ($y = 0$) for many combinations of laser power and scan speed corresponding to those experimentally examined as listed in Table 1. The melt pool dimensions estimated with the model were compared to those experimentally measured from optical microscopy. Properties of 316L SS employed for the estimation were: density ($\rho$) = 7800 kg/m$^3$; specific heat, $c_p$ = 460 J/kgK; thermal conductivity, k = 14 W/mK; absorption, A = 0.35; thermal diffusivity, a = 3.9 × 10$^{-6}$; melting point, $T_m$ = 1678 K.

## 4. Results and Discussion

### 4.1. Starting 316L Stainless Steel Powders

The gas-atomized 316L SS powders were spherical in shape with occasional satellites as presented in Figure 1a. The cross-sectional backscatter electron micrograph shown in Figure 1b clearly revealed the rapidly solidified dendritic microstructure. The particle size in Figure 1c exhibited a Gaussian-type distribution. The measured D10, D50 and D90 values were, respectively, 22.0 μm, 35.5 μm and 50.0 μm. XRD pattern in Figure 1d confirmed the presence of the FCC austenite phase, without any detectable presence of the ferrite phase. The powder composition from the SEM-XEDS measurement is reported in Table 2, and is close to the specification published by SLM. (Solutions Group AG, Lübeck, Germany).

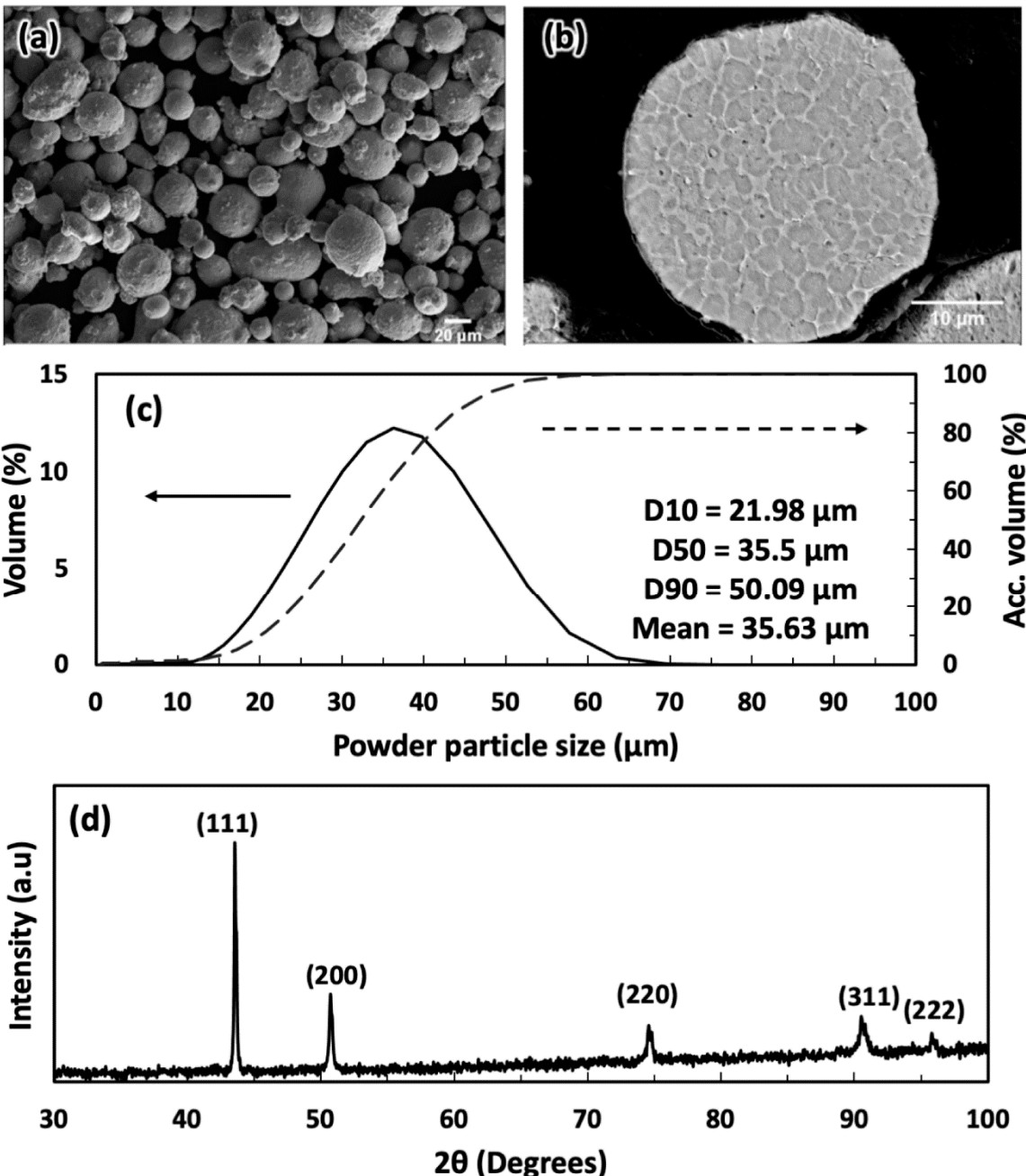

**Figure 1.** Characteristics of the gas atomized 316L SS powders examined by (**a**) secondary electron micrograph; (**b**) cross-sectional backscatter electron micrograph; (**c**) powder size distribution; and (**d**) X-ray diffraction pattern.

**Table 2.** Chemical composition of the gas atomized 316L SS powders determined by SEM-XEDS and the nominal composition specification from SLM Solutions Group AG (Lübeck, Germany).

| Method | Si | Cr | Mn | Fe | Ni | Mo |
|---|---|---|---|---|---|---|
| SEM-XEDS | $0.7 \pm 0.1$ | $18.5 \pm 0.2$ | $1.8 \pm 0.2$ | $68.3 \pm 0.3$ | $8.9 \pm 0.1$ | $1.8 \pm 0.1$ |
| SLM Specification | max 1.0 | 16.0–18.0 | max 2.0 | BAL. | 10.0–14.0 | 2.0–3.0 |

### 4.2. Influence of LPBF Parameters on the Density

Figure 2 presents representative optical micrographs of samples examined (series I in Table 1) and quantified using image analysis. Dark contrast features correspond to pores and flaws in the as-built 316L SS. The dotted region in Figure 2 with energy density values between 45 and 125 J/mm$^3$ produced samples with density greater than 99.8% determined by image analysis. This optimized energy density range includes the optimum energy density of ~100 J/mm$^3$ reported in the literature [39–41].

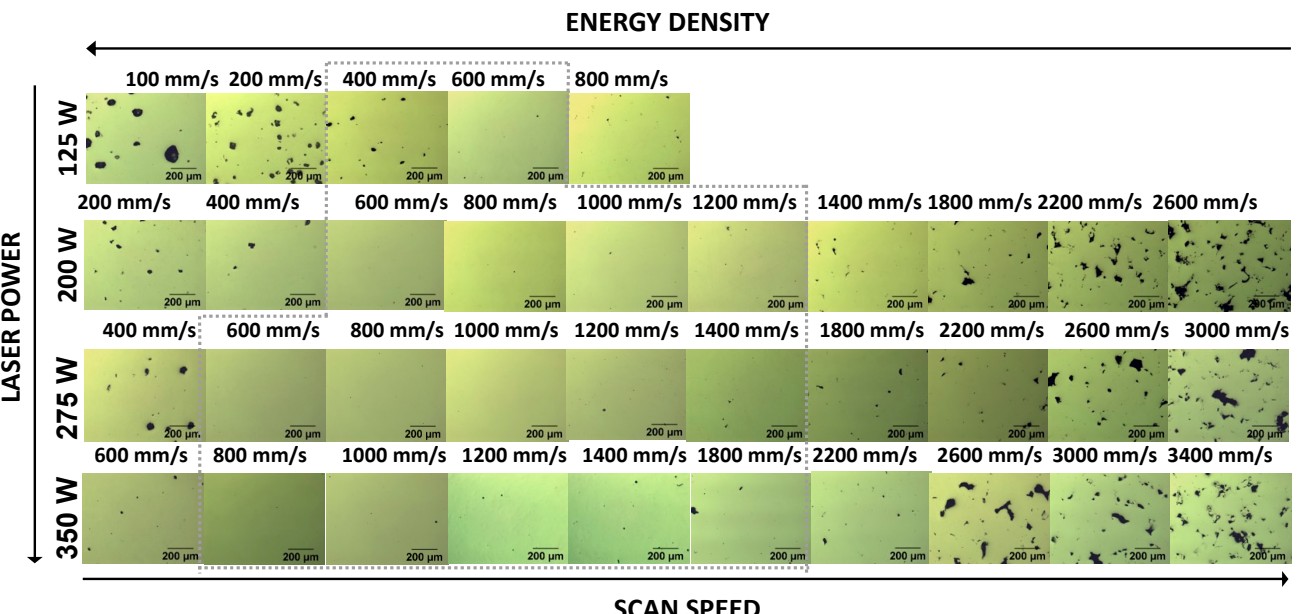

**Figure 2.** Optical micrographs from the cross-sections parallel to the build direction for LPBF 316L SS (Serie I) built as functions of laser power and scan speeds. Hatch spacing and slice thickness were kept constant at 0.12 mm and 0.03 mm, respectively.

Lower volumetric energy density, for example, with high laser scan speed at a constant laser power yielded more irregular-shaped flaws due to insufficient melting, i.e., lack of fusion flaws. Higher energy density on the other hand produced more rounded pores, which can be attributed to the keyhole effect. The presence of flaws and/or pores decreased significantly for a range of intermediate volumetric energy density. To quantify these observations, an extensive image analysis was carried out.

Figure 3a presents the relative density determined from samples produced by LPBF as function of scan speed for various laser power employed as listed in Table 1 (series I). In general, relative density increased sharply with an increase in scan speed, remained above 99.8%, then decreased gradually with a further increase in scan speed. Density higher than 99.8% was observed for the sample produced with the scan speed from 600 mm/s to 1200 mm/s, from 600 mm/s to 1400 mm/s, and from 800 mm/s to 1400 mm/s for the laser power of 200 W, 275 W and 350 W, respectively.

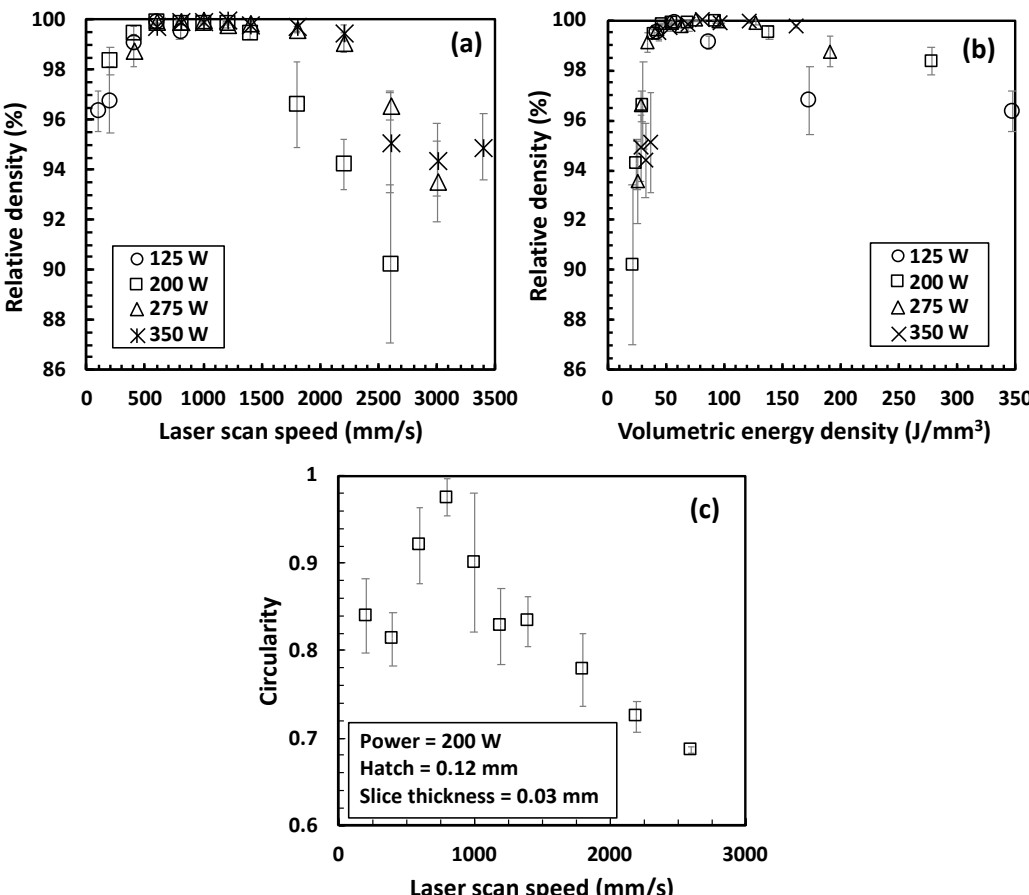

**Figure 3.** Relative density as a function of (**a**) laser scan speed and (**b**) volumetric energy density; (**c**) Circularity of flaws as a function of laser scan speed with the laser power, hatch distance, and slice thickness held constant at 200 W, 0.12 mm, and 0.03 mm, respectively.

The same results from density measurement can be presented as a function of volumetric energy density, as plotted in Figure 3b. The density of the samples sharply increased from ~90% with an increase in volumetric energy density, and reached a density greater than 99.8% around 46 J/mm$^3$. The density remained greater than 99.8% from 46 J/mm$^3$ to 127 J/mm$^3$, but decreased gradually down to ~95% with a further increase in volumetric energy density, as presented in Figure 3b. Within the LPBF parameters examined in this study, using constant hatch spacing and slice thickness of 0.12 mm and 0.03 mm, respectively, the density variation could be described well by the volumetric energy density variation.

Figure 3c presents the circularity of pores and flaws as a function of scan speed for a laser power of 200 W. Circular pores with circularity near 1 would more likely correspond to keyhole pores, while circularity much lower than 1 would correspond to flaws originating from insufficient melting, i.e., lack-of-fusion flaws from interparticle space residuals. At very low scan speeds, where keyhole pores may develop, flaws with slightly lower circularity were observed, perhaps due to an incomplete coalescence of multiple keyhole pores. At very high scan speeds, where lack-of-fusion flaws may develop, flaws with lower circularity were observed. Between 600 mm/s and 1000 mm/s of scan speed, circularity higher than 0.9 was observed for the laser power of 200 W employed. This trend in circularity was similar for other laser powers employed. The LPBF parameter specified by SLM Solutions, (power = 200 W; scan speed = 800 mm/s; hatch spacing = 0.12 mm; slice thickness = 0.03; energy density = 69.4 J/mm$^3$) corresponded to the very high density (>99.8%) and very high circularity (>0.95). Therefore, this corresponds to the optimum LPBF parameter set determined in this study.

To examine the effect of hatch distance, LPBF of 316L SS was carried out using 0.08 mm, 0.1 mm, 0.12 mm, 0.14 mm and 0.16 mm, while the laser power, scan speed, and slice thickness were held constant at 200 W, 800 mm/s, and 0.03 mm, respectively. This investigation corresponds to series II in Table 1. In general, as the hatch distance decreased, the density increased, as presented in Figure 4a, although the relative density remained mostly greater than 99.8%. This would be attributed to the fact that even at the large hatch distance of 0.16 mm, which is much larger than the powder size or laser beam diameter, the melt pools are sufficiently large, and they overlap to minimize the "lack-of-fusion" flaws. Still, with an increase in hatch distance, the circularity of the pores decreased, as demonstrated in Figure 4b, which indicates the progressively increasing formation of "lack-of-fusion" flaws.

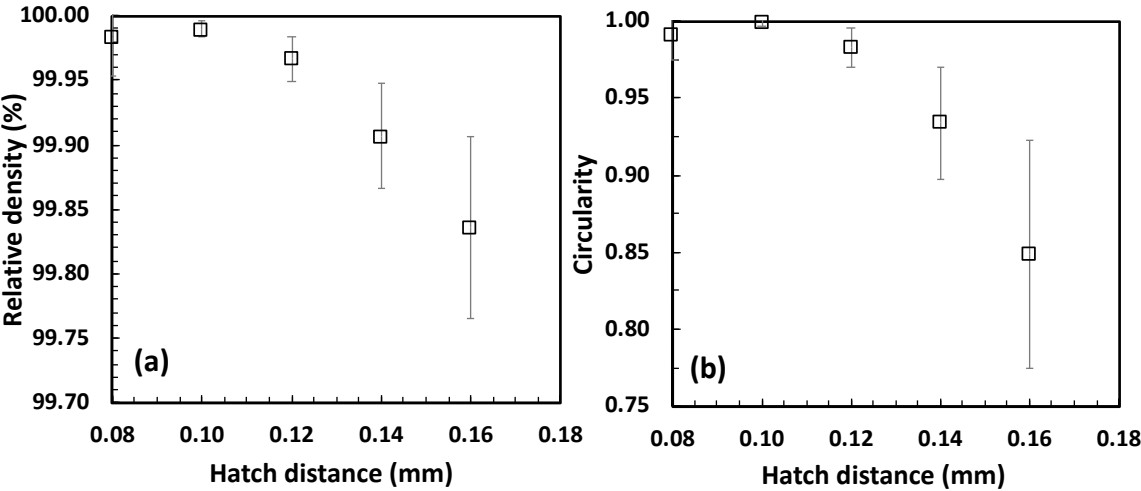

**Figure 4.** (**a**) Relative density and (**b**) flaw circularity as a function of hatch distance with the laser power, scan speed, and slice thickness held constant at 200 W, 800 mm/s, and 0.03 mm, respectively.

### 4.3. Influence of LPBF Parameters on the Microstructure

Figure 5 presents characteristic microstructural features of the LPBF-built 316L SS. Parallel to the build direction, a typical melt pool feature was observed as presented in Figure 5a. Perpendicular to the build direction, discontinuous melt pool tracks were observed as shown in Figure 5b, due to the scan rotation of 67° employed in this study. Columnar grains orthogonal to the melt pool boundaries within individual melt pools were observed as presented in Figure 5a. Within the melt pools and within the columnar grains, fine cellular and columnar-cellular structures were observed as presented in Figure 5c,d. The difference in aspect ratio observed for these cells would be due to the variation of column orientations observed in 2-dimensional micrographs.

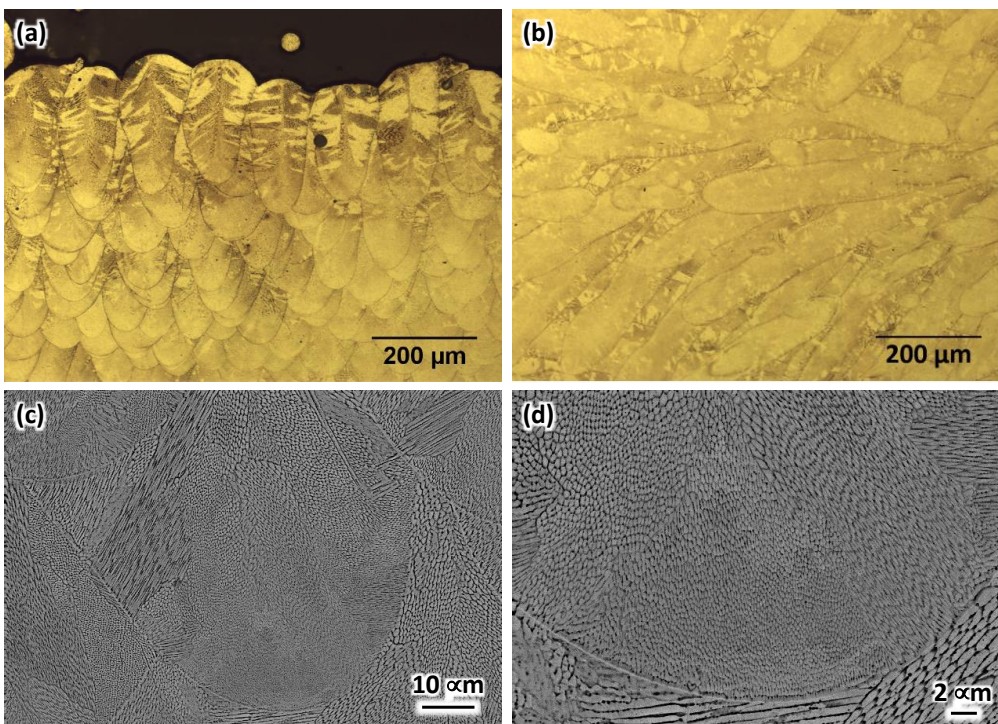

**Figure 5.** Optical micrographs of the LPBF 316L SS (**a**) parallel to the build direction and (**b**) perpendicular to the build direction. Backscattering electron micrographs parallel to the build direction at (**c**) low and (**d**) high magnifications. These samples were produced with a laser power, scan speed, hatch spacing, and slice thickness of 275 W, 1000 mm/s, 0.12 mm, and 0.03 mm, respectively.

Melt pool depth and width were measured from the very top layer of the melt pools parallel to the build direction. Figure 6 presents typical melt pools observed from optical microscopy for samples produced as a function of scan speed at laser power of 275 W, hatch spacing of 0.12 mm, and slice thickness of 0.03 mm. These melt pools were only exposed to a single laser scan (i.e., last scan), and symmetry of the melt pool shape was assumed for the measurement, as indicated in Figure 6. Table 3 and Figure 7 present experimentally measured depth and width of melt pools in this study. It should be noted that melt pool width and depth could not be determined with confidence when the laser scan speed was exceedingly high, e.g., 3000 mm/s at 275 W and 3000 and 3500 at 350 W, so they are not reported in Table 3 and Figure 7.

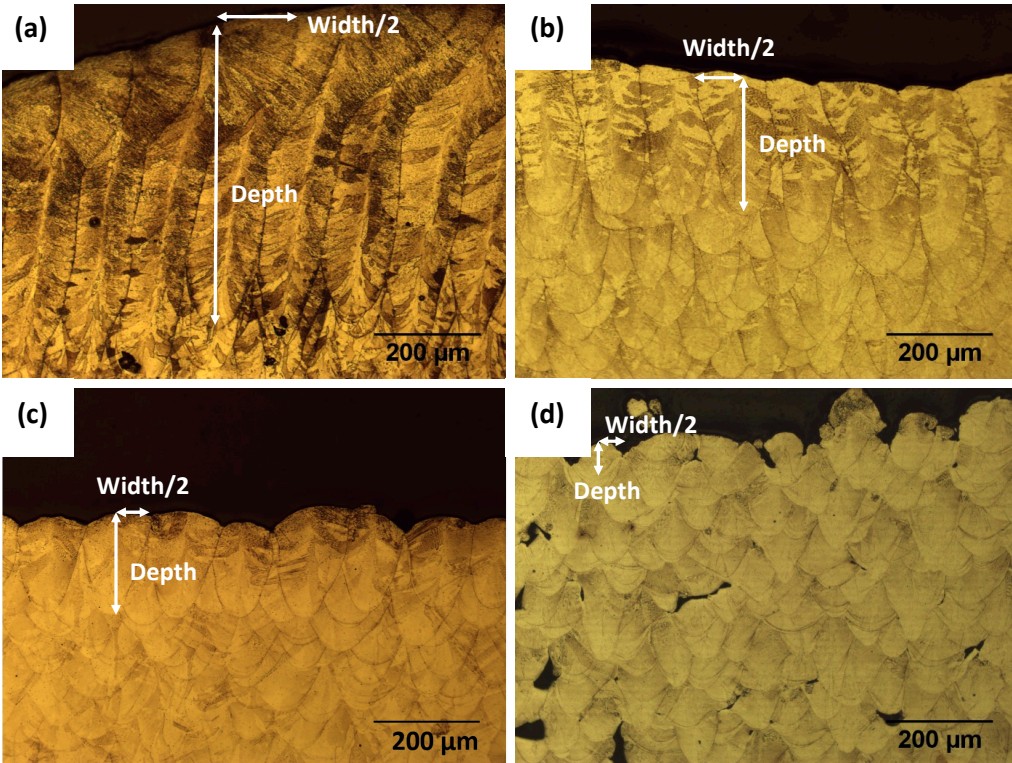

**Figure 6.** Optical micrographs of melt pools perpendicular to the build direction from the last top layer solidified in 316L SS samples as a function of scan speed: (**a**) 400 mm/s, (**b**) 800 mm/s, (**c**) 1200 mm/s, and (**d**) 2200 mm/s. Laser power, hatch spacing, and slice thickness were held constant at 275 W, 0.12 mm, and 0.03 mm, respectively.

**Table 3.** Melt pool width and depth determined from experimental measurements and estimated from analytical Rosenthal solution.

| Power (W) | Scan Speed (mm/s) | Slice Thickness (mm) | Hatch Distance (mm) | Volume Energy Density (J/mm³) | Melt Pool Depth exp * (μm) | Melt Pool Depth cal ** (μm) | Melt Pool Width exp * (μm) | Melt Pool Width cal ** (μm) |
|---|---|---|---|---|---|---|---|---|
| 125 | 100 | 0.03 | 0.12 | 347.22 | 481 ± 56 | 134 | 254 ± 106 | 268 |
| 125 | 200 | 0.03 | 0.12 | 173.61 | 274 ± 47 | 99 | 210 ± 57 | 198 |
| 125 | 400 | 0.03 | 0.12 | 86.81 | 163 ± 42 | 72 | 151 ± 28 | 144 |
| 125 | 600 | 0.03 | 0.12 | 57.87 | 78 ± 14 | 60 | 120 ± 19 | 120 |
| 125 | 800 | 0.03 | 0.12 | 43.40 | 52 ±12 | 52 | 111 ± 14 | 104 |
| 200 | 200 | 0.03 | 0.12 | 277.78 | 546 ±67 | 128 | 309 ± 77 | 256 |
| 200 | 400 | 0.03 | 0.12 | 138.89 | 368 ± 27 | 92 | 234 ± 35 | 184 |
| 200 | 600 | 0.03 | 0.12 | 92.59 | 268 ± 38 | 76 | 222 ± 41 | 152 |
| 200 | 800 | 0.03 | 0.12 | 69.44 | 169 ± 18 | 66 | 148 ± 19 | 132 |
| 200 | 1000 | 0.03 | 0.12 | 55.56 | 115 ± 25 | 59 | 142 ± 15 | 118 |
| 200 | 1200 | 0.03 | 0.12 | 46.30 | 91 ± 15 | 54 | 114 ± 18 | 108 |
| 200 | 1400 | 0.03 | 0.12 | 39.68 | 62 ± 18 | 50 | 99 ± 22 | 100 |
| 200 | 1800 | 0.03 | 0.12 | 30.86 | 71 ± 30 | 44 | 97 ± 26 | 88 |
| 200 | 2200 | 0.03 | 0.12 | 25.25 | 27 ± 11 | 40 | 63 ± 15 | 80 |
| 200 | 2600 | 0.03 | 0.12 | 21.37 | 52 ± 16 | 37 | 94 ± 22 | 74 |
| 275 | 400 | 0.03 | 0.12 | 190.97 | 590 ± 47 | 109 | 318 ± 140 | 218 |
| 275 | 600 | 0.03 | 0.12 | 127.32 | 394 ± 24 | 89 | 195 ± 29 | 178 |
| 275 | 800 | 0.03 | 0.12 | 95.49 | 290 ± 28 | 77 | 182 ± 31 | 154 |
| 275 | 1000 | 0.03 | 0.12 | 76.39 | 205 ± 27 | 69 | 125 ± 25 | 138 |
| 275 | 1200 | 0.03 | 0.12 | 63.66 | 148 ± 20 | 64 | 130 ± 22 | 128 |
| 275 | 1400 | 0.03 | 0.12 | 54.56 | 98 ± 29 | 59 | 105 ± 18 | 118 |
| 275 | 1800 | 0.03 | 0.12 | 42.44 | 78 ± 25 | 51 | 88 ± 16 | 102 |
| 275 | 2200 | 0.03 | 0.12 | 34.72 | 61 ± 25 | 47 | 68 ± 17 | 94 |
| 275 | 2600 | 0.03 | 0.12 | 29.38 | 53 ± 15 | 43 | 78 ± 19 | 86 |
| 350 | 600 | 0.03 | 0.12 | 162.04 | 605 ± 35 | 89 | 280 ± 125 | 174 |
| 350 | 800 | 0.03 | 0.12 | 121.53 | 409 ± 14 | 77 | 223 ± 57 | 148 |
| 350 | 1000 | 0.03 | 0.12 | 97.22 | 322 ± 30 | 69 | 218 ± 45 | 134 |
| 350 | 1200 | 0.03 | 0.12 | 81.02 | 209 ± 31 | 64 | 138 ± 73 | 124 |
| 350 | 1400 | 0.03 | 0.12 | 69.44 | 152 ± 47 | 59 | 151 ± 39 | 116 |
| 350 | 1800 | 0.03 | 0.12 | 54.01 | 115 ± 34 | 52 | 99 ± 18 | 104 |
| 350 | 2200 | 0.03 | 0.12 | 44.19 | 70 ± 30 | 47 | 87 ± 18 | 90 |
| 350 | 2600 | 0.03 | 0.12 | 37.39 | 81 ± 17 | 43 | 116 ± 21 | 86 |

Note: exp * refers to experimental measurement and cal ** refers to calculated based on analytical Rosenthal solution. These are reported with average ± standard deviation values.

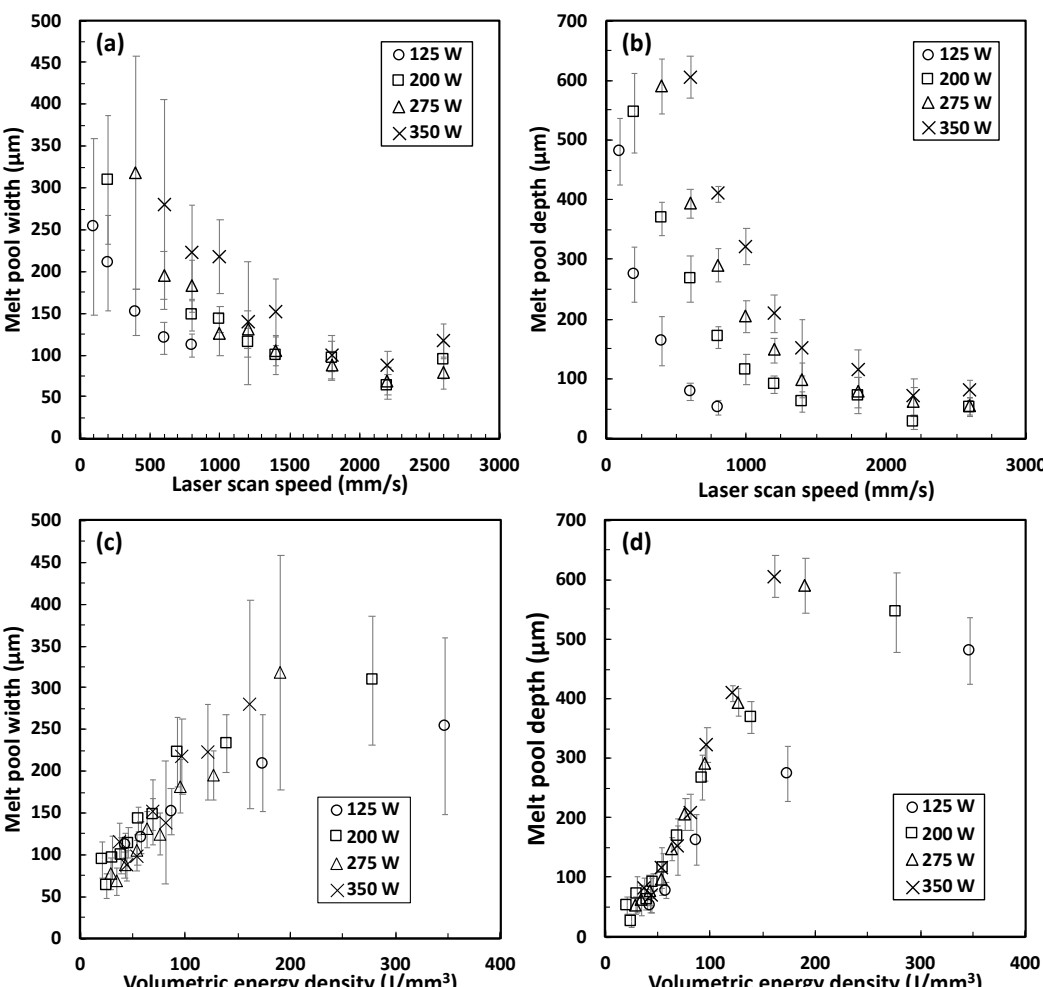

**Figure 7.** Variation of melt pool (**a**) width and (**b**) depth as a function of laser scan speed; and variation of melt pool (**c**) width and (**d**) depth as a function of volumetric energy density.

Variation in melt pool width and depth as a function of laser scan speed is presented in Figure 7a,b, respectively, for the laser powers employed in this study. Clearly, both the width and depth decrease with an increase in scan speed and a decrease in power. The same result can be plotted as a function of volumetric energy density, as shown in Figure 7c,d. Both the width and depth increase with an increase in volumetric energy density up to approximately 200 J/mm³. However, this trend does not hold, albeit only a few data points, with excessive volumetric energy density above 200 J/mm³. However, a close examination reveals that the increases in width and depth with volumetric energy density is consistent for a fixed laser power. Therefore, the dispersion in data observed in Figure 7c,d at higher volumetric energy density is a result due to change in laser power (at extremely slow scan speed) that cannot be normalized by volumetric energy density.

The melt pool dimensions examined as functions of LPBF parameters (e.g., power and scan speed) and volumetric energy density also demonstrated that, in terms of absolute scale, the changes in depth were more sensitive than width. For example, the width measured in this study, on average, ranged from 50 to 300 m, while the depth measured ranged from 30 to 600 m.

Tang et al. [59] formulated a relationship between the occurrence of lack-of-fusion to the melt pool dimensions by introducing the criterion expressed by:

$$\left(\frac{h}{w}\right) + \left(\frac{t}{d}\right) \leq 1 \tag{5}$$

where h is the hatch spacing, w the melt-pool width, t the layer/slice thickness, and d the melt-pool depth. In the case where the criterion is more than 1, incomplete melting occurs, and the lack-of-fusion flaw is predicted. For the case less than 1, lack-of-fusion flaws are avoided due to overlapping melting. Figure 8 shows the calculated criterion based on experimental measured melt pool dimension as a function of volumetric energy density for the 316L SS samples. A rapid decrease in the value for the criterion of lack-of-fusion is observed when volumetric energy density increased. The volumetric energy density range (highlighted in gray box from 46 J/mm$^3$ to 127 J/mm$^3$) that yielded samples with greater than 99.8% relative density closely corresponded to the criterion below 1, except for the lower end, e.g., <55 J/mm$^3$.

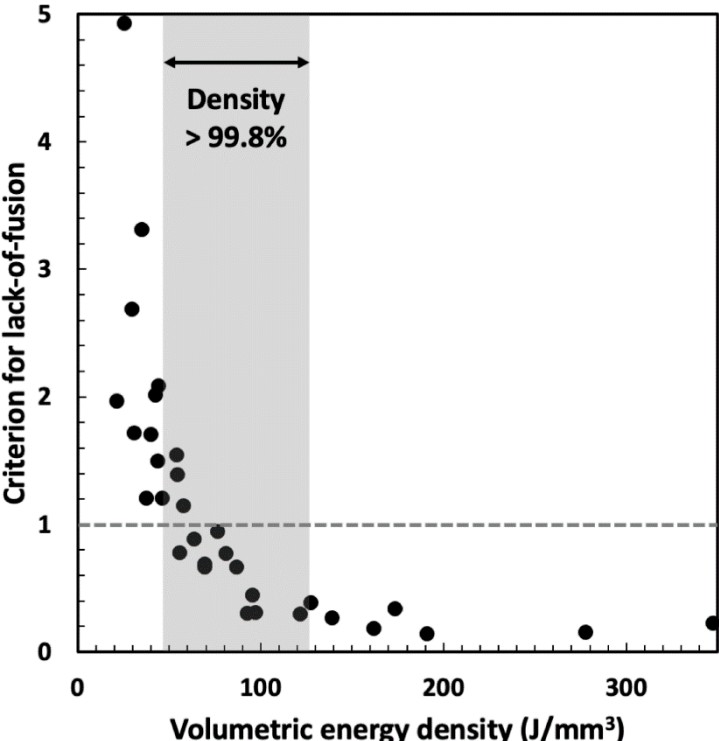

**Figure 8.** Criterion for lack-of-fusion determined using Equation (5) plotted as a function of energy density. Gray box corresponds to the volumetric energy density that yielded relative density greater than 99.8%.

The size of cellular structure presented in Figure 5c,d reflects the cooling rate ($\dot{T}$) during solidification. For 316L SS, the relationship between secondary dendrite arm spacing ($\lambda$) and cooling rate has been described as [60]:

$$\lambda = 80\dot{T}^{-0.33} \tag{6}$$

Substituting the experimentally observed cell size of ~0.4 to 0.8 μm yield cooling rate of ~$10^5$ to $10^7$ K/s is consistent with that of other studies [15,45,53–56]. Unfortunately, detailed analysis as functions of LPBF parameters was not fruitful due to large standard deviation (from melt pool to melt pool and due to orientation variation) in measured cell size.

### 4.4. Melt Pool Dimension Estimated by Rosenthal Solution

Rosenthal solutions, originally designed for welding processes, have found application in LPBF studies, because it is a very simple analytical method to predict/analyze the melt pool development. In this study, the melt-pool depth and width were estimated with Equation (2) with absorption and alloy properties listed in Section 3, however with an

assumption that the melt pool depth is half of the melt pool width. Figure 9 illustrates representative temperature contour estimated from the top surface as the laser scan is carried out. The shape, both the width and length, of the melt pool estimated by yellow corresponding to the melting temperature (T = 1678 K) is strongly dependent on the LPBF parameters.

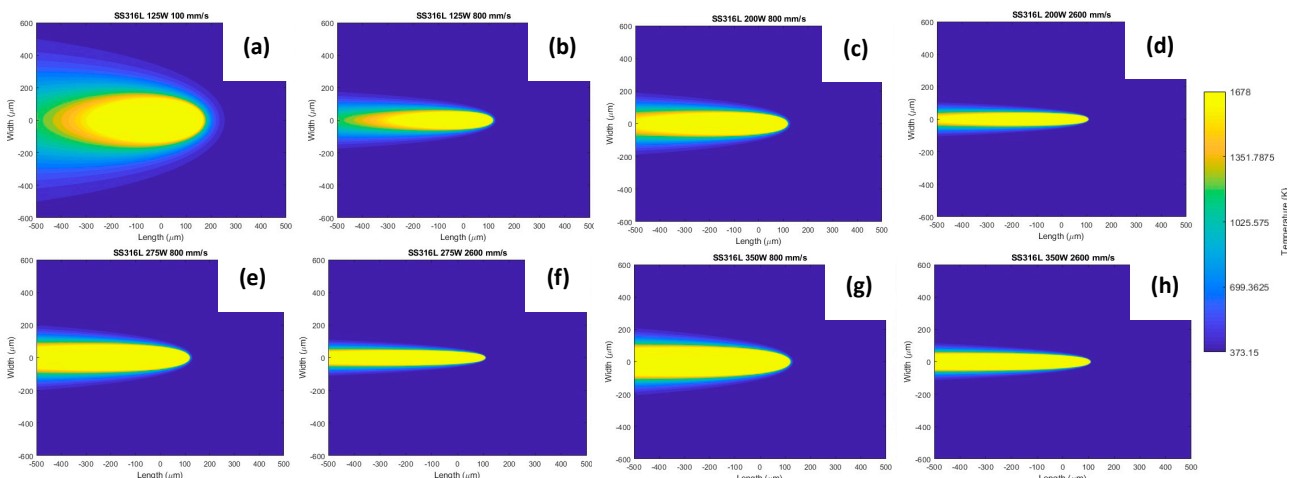

**Figure 9.** Characteristic temperature distributions calculated from Rosenthal's solution for: (**a**) 125 W at 100 mm/s, (**b**) 125 W at 800 mm/s, (**c**) 200 W at 800 mm/s and (**d**) 200 W at 2600 mm/s, (**e**) 275 W at 800 mm/s and (**f**) 275 W at 2600 mm/s, (**g**) 350 W at 800 mm/s and (**h**) 350 W at 2600 mm/s.

Table 3 presents the comparison between the experimentally measured and calculated melt pool width and depth. Figure 10 presents a representative comparison, for example, when the laser power employed was 200 W. Both width and depth increased as the scan speed decreased, but the calculated values are always lower than the experimental results, especially at low scan speed, i.e., higher volumetric energy density and for melt pool depth. Therefore, to fully utilize the simplicity of Rosenthal solutions, temperature dependent thermo-physical properties, energy density dependent absorption, and depth due to keyholing [44], among others should be investigated in detail to better model the temperature distribution and melt pool development [61–63].

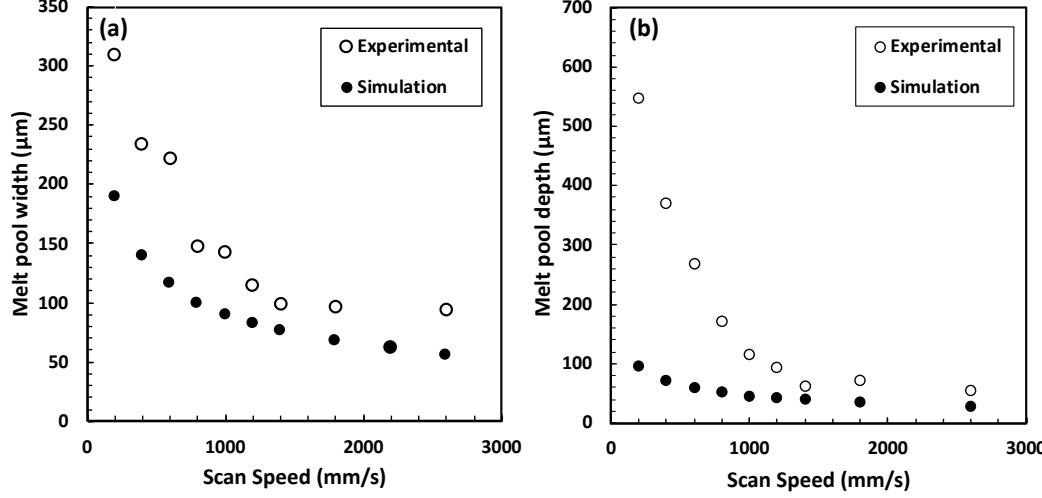

**Figure 10.** Melt pool (**a**) width and (**b**) depth determined by experimental measurements and estimated by Rosenthal solution when the laser power, hatch spacing and slice thickness employed were 200 W, 0.12 mm and 0.03 mm, respectively.

Figure 11 presents engineering stress vs. engineering strain responses of three tensile specimens built with an optimized set of parameters: 200 W laser power, 800 mm/s scan speed, 0.12 mm hatch spacing and 0.03 mm slice thickness, corresponding to a volumetric energy density of ~69 J/mm$^3$. Tensile properties such as yield strength (YS), Young's modulus (GPa), ultimate tensile strength (UTS), and strain at failure extracted are reported in Table 4. These properties are similar to 316L SS produced by a conventional method, and to others [41,45–48] reported for as-built LPBF 316L SS.

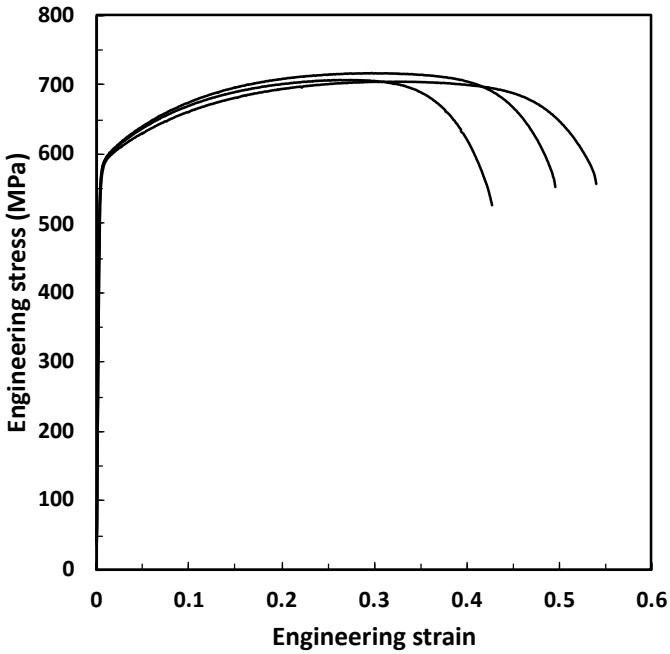

**Figure 11.** Tensile stress-strain curves for LPBF 316L SS specimens 3 built with laser power of 200 W, scan speed of 800 mm/s, hatch spacing of 0.12 mm and slice thickness of 0.03 mm.

**Table 4.** Tensile properties of the LPBF 316L SS printed at the optimized parameter determined from this study.

| Sample | YS (MPa) | E (GPa) | UTS (MPa) | Strain at Failure (%) |
|---|---|---|---|---|
| 316L SS (#1) | 558.2 | 182.9 | 705.1 | 54.0 |
| 316L SS (#2) | 567.2 | 187.4 | 707.3 | 42.8 |
| 316L SS (#3) | 564.4 | 165.9 | 717.4 | 49.7 |
| Average and Standard Deviation | 563.3 ± 3.8 | 178.7 ± 9.3 | 709.9 ± 6.6 | 48.3 ± 5.6 |

Wang et al. [45] reported an empirical relationship between yield strength and sub-grain cell size as, $\sigma_y = \sigma_0 + k_{HP}/\sqrt{PDAS} = 183.31 + \left(253.66/\sqrt{\lambda}\right)$, based on the assumption of Hall–Petch-type strengthening behavior. The smaller cell size observed in this study, ~0.4 mm, corresponds to the yield strength reported in Table 4, and therefore additional factors that contribute to the strength need to be identified. The fracture surfaces of the tensile specimens were examined by using SEM and they are shown in Figure 12. All samples failed in a fairly ductile mode as indicated by the presence of fine dimples on the fracture surfaces, despite the presence of several voids on the facture surfaces.

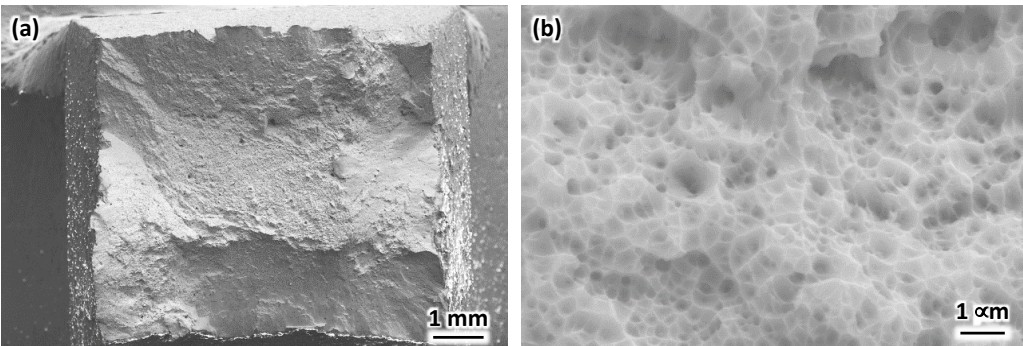

**Figure 12.** (**a**) Low and (**b**) high resolution secondary electron micrographs from the fracture surface of LPBF 316L SS built at 200 W laser power, 800 mm/s scan speed, 0.12 mm hatch spacing and 0.03 mm slice thickness.

### 5. Conclusions

This study was designed to investigate the effect of processing parameters on the densification, melt pool morphology and microstructural evolution of LPBF SS 316L by the use of volumetric energy density, and an analytical model based on Rosenthal solution. Laser power and scan speed were varied over a wide range, systematically and independently.

Key findings of this investigation are:

- The energy density input affects the overall pores and flaws observed in LPBF 316L SS. Volumetric energy density, below 46 J/mm$^3$, yielded "lack-of-fusion" flaws due to insufficient melting, while excessive energy density, above 127 J/mm$^3$, produced "keyhole" porosity. Between these two extremes, there was a wide range of volumetric energy density in which density greater than 99.8% was achieved.
- Width and depth of melt pool increased with higher volumetric energy density (e.g., higher power and slower scan speed). Variation in melt pool width and depth as a function of energy input was calculated using a simple Rosenthal solution, and compared to experimental measurements.
- The threshold for lack of fusion can be used to help identify the onset of optimum LPBF parameters which would yield high density alloy specimens/components.
- As-built microstructure in LPBF 316L SS consisted of sub-grain cellular structures within grains observed normal to the boundaries of the melt pool structure. Cooling rate was estimated to be around 10$^5$ to 10$^7$ K/s based on the size of these cells.
- Consistent as-built mechanical properties, YS = 563 MPa, E = 179 GPa, UTS = 710 MPa, and elongation at fracture = 48% was observed for the sample build with volumetric energy density of 69 J/mm$^3$. These properties were correlated to a relative density greater than 99.8% and cell size of ~0.4 μm. The predominant mode of fracture was ductile.

**Author Contributions:** Conceptualization, N.D.V. and Y.S.; formal analysis, N.D.V. and K.G.; investigation, N.D.V., C.L. and N.A.; methodology, N.D.V. and H.H.; writing—original draft, N.D.V. and Y.S.; writing—review and editing, N.D.V. and Y.S.; supervision, Y.S.; project administration, Y.S. funding acquisition, Y.S. All authors have read and agreed to the published version of the manuscript.

**Funding:** This research was sponsored by the DEVCOM U.S. Army Research Laboratory under a cooperation agreement contract, W911NF1720172. The views, opinions and conclusions made in this document are those of the authors and should not be interpreted as representing the official policies, either expressed or implied, of the DEVCOM U.S. Army Research Laboratory or the U.S. Government. The U.S. Government is authorized to reproduce and distribute reprints for Government purposes notwithstanding any copyright notation herein. Nathalia Diaz Vallejo expresses her gratitude to the Fulbright Program and the Colombian Institute of Educational Credit and Technical Studies (ICETEX) for providing partial stipend for her doctoral studies under the "Fulbright-Pasaporte a la Ciencia" fellowship program.

**Institutional Review Board Statement:** Not applicable.

**Informed Consent Statement:** Not applicable.

**Data Availability Statement:** The data presented in this study are available on request from the corresponding author.

**Conflicts of Interest:** The authors declare no conflict of interest.

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
