# Peer review of "Process Optimization and Microstructure Analysis to Understand Laser Powder Bed Fusion of 316L Stainless Steel"

_metals, doi:10.3390/met11050832_

Round 1

Reviewer 1 Report

Dear Authors,

The reviewed submission titled: “Process optimization and microstructure analysis to understand laser powder bed fusion of 316L stainless steel “ is an experimental work on additive manufacturing of 316L austenitic stainless steel. The work has a classic IMRaD layout, which perfectly corresponds to the content presented in it. The subject of the manuscript is in line with the current research trends in the processing of metal alloys and is in line with the scope of the Metals journal. In my opinion, the manuscript can be published, but it requires a few important additions and explanations, which I provide below:

Keywords: correct: "316L stainless steel"

Throughout the text, use a uniform wording: "316L SS".

Introduction:

Please use a citation method consistent with the journal's guidelines: [1-6], [10,11,12]. Use the notation: "Figure" instead of "Fig."

My general impression after reading this chapter is that it is too short to introduce the reader to the research part well and justify taking up the topic. In the literature, you can find many current articles describing the testing of weldability, mechanical and corrosion properties of austenitic stainless steels. In this regard, please consider relying on information from the works: https://doi.org/10.3390/ma13204540, https://doi.org/10.3390/met10050559

In my opinion, this section lacks a deeper analysis of the state of the art. Only basic information directly related to the research part of the work is included. It would be valuable to highlight the material aspect of austenitic steels, which is only mentioned in lines: 39-42.

Line 55: Add the source of the equation.

Please clearly state the purpose of the work. Lines: 68-76: this is the work scope that should be described in chapter 2.

Chapter 2:

On what basis were the process parameters selected? Are they assumed on the basis of previous experience or literature information?

Were the samples etched?

Please provide the names of all devices used for testing (welding machine as well), names of manufacturers and their addresses (according to the journal guidelines).

Table 1: is the measurement error shown in the last column the standard deviation?

Line 149: "Starting" does not seem to be the correct term. Maybe a better is: "initial state"?

What error do the error bars show? Standard deviation?

Replace the unit: "sec" with "s".

Line 271: Correct a typo in the name.

Lines 278, 279: correct the unit notation.

Line 366: Correct typos in figure caption.

Chapter 5 title: I suggest replacing the title: "Summary" with "Conclusions". Please add the first sentence describing the subject (main idea) of your research.

Section is missing: Authors Contributions.

References must be formatted in accordance with the journal's guidelines. I propose to add current articles from the MDPI publishing house.

Reviewer 2 Report

The different effects of process parameters on densification and microstructure of SS 316L are investigated. Melt-pool dimensions are studied by experiments and analytical Rosenthal solution. The subject is interesting, however, some revisions should be done prior to acceptance:

  1. English should be substantially improved. In the current state, the manuscript is very hard to follow. There are many spelling and grammatical errors in the manuscript e.g. (Line 253: lase).
  2. The formula between Line 127 and Line 128 is Eq. (2). Please check it carefully.
  3. Introduction should be almost re-written. The Rosenthal solution model was developed in the past. Introduction should discuss the state of the art in this regard, and their limitations, advantages and disadvantages should be discussed. This will clarify why the model is required, and which is the main aim and novelty of this model.
  4. The temperature dependent thermo-physical properties are important to analyze the temperature distribution during laser powder bed fusion process. In the manuscript, the properties are set as a constant. Please consider carefully.
  5. Does thermal convection need to be considered in the simulation process. What about radiative heat losses?
  6. There is a big difference between simulation and experiment results. How to ensure the correctness of the model?

Reviewer 3 Report

-Excellent work - what can I say - this is just very good
-I would re-name the "Summary" - Conclusions, especially since you presenting bullet points
-The rest is all fine with me

Round 2

Reviewer 1 Report

Dear Authors, 

thank you very much for introducing the changes suggested by me and explaining my comments. I am convinced that your article will be a good source of information for other scientists and will be gladly cited.

Author Response

Thank you for the encouraging comments. We will continue to work towards better understanding of the process with high quality experiments and refined modeling.

Reviewer 2 Report

The paper is publishable.

Author Response

(The authors gave the same response as above.)
